# Evaluation of a Community Intervention to Reduce Harmful Alcohol Consumption among Adult Population: A Study Protocol

**DOI:** 10.3390/ijerph19148746

**Published:** 2022-07-18

**Authors:** Victoria Porthé, Irene García-Subirats, Catrina Clotas, Elia Diez

**Affiliations:** 1Centro de Investigación Biomédica en Red Epidemiología y Salud Pública (CIBERESP), Avenida Monforte de Lemos, 3-5 (Pabellón 11. Planta 0), 28029 Madrid, Spain; 2Agència de Salut Pública de Barcelona, Pl. Lesseps, 1, 08023 Barcelona, Spain; igsubira@aspb.cat (I.G.-S.); cclotas@aspb.cat (C.C.); 3Institut d’Investigació Biomèdica de Sant Pau, Carrer de Sant Quintí, 77, 08041 Barcelona, Spain; 4Department of Experimental and Health Sciences, Universitat Pompeu Fabra, Doctor Aiguader 88, 08003 Barcelona, Spain

**Keywords:** harmful alcohol consumption, community participation, community-based interventions, alcohol accessibility, alcohol availability

## Abstract

Harmful alcohol consumption is shaped by a complex range of structural, social, and individual determinants that interact with inequality axes, which can be addressed at the community level. Under the framework of *Barcelona Health in the Neighborhoods*, which is a community strategy to reduce health inequalities in Barcelona’s most deprived neighborhoods, a community steering group will co-design a multicomponent community intervention. Aims: to assess its effects on: (1) alcohol accessibility, availability, and consumption at the environmental level, and (2) psychosocial and cognitive determinants of harmful alcohol consumption at the individual level. *Methods*: Quasi-experimental design with a comparison group, and pre- and post-intervention measures. Three Barcelona neighborhoods will be assigned to the intervention or comparison group based on three criteria: healthcare data on alcohol use, socioeconomic characteristics, and population size. The intervention includes activities promoting community mobilization, law enforcement, and communication campaigns in the intervened neighborhoods. Non-participant observations in standardized census sections will be performed in public spaces to collect information on three outcomes: alcohol accessibility, availability, and signs of alcohol consumption. Data collection includes a survey to a sample of 622 subjects to detect differences on these outcomes: risk awareness, knowledge, and self-efficacy about harmful alcohol consumption and drinking patterns. *Discussion*: This protocol to assess the effects of a multicomponent community intervention on harmful alcohol consumption at the environmental and population level will provide evidence on effective community health interventions and enable informed decisions for policy makers. This protocol could also be used as an implementation guide for studies aimed at reducing harmful drinking in cities with similar characteristics.

## 1. Introduction

Worldwide, 3 million deaths result every year from harmful use of alcohol, which represents 5.3% of all deaths and 5.1% of the global burden of disease and injury [1]. In 2010, WHO developed the *Global strategy to reduce the harmful use of alcohol*, with the aim of reducing the harmful use of alcohol as a public health priority [2]. Since 1992, the WHO European region has been carrying out action plans aimed at reducing alcohol-related harm [3], and in 2020, a new Action Plan (2022–2030) has been developed to contribute to the effective implementation of the global strategy to reduce the harmful use of alcohol.

Alcohol use and its related problems are shaped by a wide and complex range of structural, intermediate, and individual determinants that interact, in turn, with inequality axes such as social class, economic status, education, gender, migrant status, or place of residence [4,5]. Individual knowledge, attitudes, skills, and personality characteristics affect motivation to engage in risky behaviors [6], but the social and cultural contexts are the determinants which play the most relevant role in framing health behaviors, including alcohol consumption [7,8]. For example, in countries where gender roles are more polarized, women tend to drink less than men [9]. At the local and community level, environmental associations appear between the availability of alcohol in neighborhoods and individual alcohol consumption [10]. High exposure to alcohol abuse in communities where binge drinking is socially accepted increases individual risk [8], and alcohol use disorders are more frequent in neighborhoods with high social isolation (low-average household occupancy) [11]. In contrast, community cohesion and high levels of social support lead to better social and health outcomes related to alcohol use [8]. Involvement in community health initiatives is associated with good results in health, especially among disadvantaged groups, partly when interventions are tailored to local needs [12,13]. There is a gap in the knowledge of community interventions to reduce alcohol use among adults. Previous evidence has shown that most of the studies reported methodological weaknesses, mainly regarding their design. Few details of the interventions’ implementation process were offered, making it more difficult to determine whether the lack of success was due to the intervention itself or to failures in the implementation. The complexity of community interventions has been described as multi-component interventions increases the complexity to evaluate and may result in an inability to detect potential effects. However, there is evidence that a small number of local and comprehensive community-based programs have been effective in preventing alcohol consumption and related problems [14,15]. 

In 2007, the Barcelona Public Health Agency, the City Council, and the Consortium of Health Services launched a community strategy to reduce health inequalities, *Barcelona Salut als Barris* (Barcelona Health in the Neighborhoods) [16]. This strategy includes: (1) the setting of community teams in the most deprived neighborhoods, (2) carrying out periodically community health diagnosis and public prioritizations, (3) reviewing the existing evidence to achieve shared objectives, and (4) co-creation, implementation, and evaluation of interventions with the community. In the community health diagnoses, problematic alcohol consumption was identified as one of the first health priorities of several neighborhoods. Therefore, we first reviewed the scientific literature to identify effective community interventions aimed at reducing problematic alcohol consumption in the adult population [17], which is less studied than the adolescent population. The review identified three common components among the effective interventions at the community setting: community mobilization, law enforcement, and media campaigns. They combined individual and environmental approaches addressing structural and cultural determinants of consumption. These results were consistent with evidence that describes how changing the social norms that encourage and trivialize the drinking environment [7,8], together with law enforcement, are effective strategies to reduce harmful alcohol consumption [18,19]. 

To address harmful alcohol consumption, we will co-design with local groups a community intervention aimed at reducing, preventing, and mitigating the social impact of problematic alcohol consumption in three disadvantaged neighborhoods in Barcelona. The intervention design is based on Bandura’s social learning theory that underpins social cognitive theory [20,21]. Social cognitive theory holds that personal, environmental, and behavioral factors influence each other’s decision-making. Personal factors, such as knowledge and attitudes, affect how a person interprets the environment, and this in turn modifies the resulting behaviors. Therefore, individual behavioral change can be achieved by altering the physical and social environment. Changes in any component—personal, behavioral, or environmental—also trigger changes in others, which, through the socialization processes, play a key role in setting individual behavioral parameters [22].

### 1.1. Hypothesis

This multi-component community intervention will lead to significant changes: Firstly, at the environmental level, in the intervened neighborhoods there will be a decrease in the accessibility and availability of alcohol, less exposure to alcohol-related advertising, and better compliance with current regulations compared with non-intervened neighborhoods. Secondly, at the individual level, the population’s knowledge, self-efficacy, and critical thinking about harmful drinking will increase.

### 1.2. General Objectives

To assess the effects of a multi-component intervention on alcohol availability, promotion, and consumption at the environmental level and in psychosocial and cognitive determinants of harmful alcohol consumption at population level. 

### 1.3. Specific Objectives

Increase knowledge of harmful alcohol consumption limits.Reduce attitudes that favor problematic alcohol consumption.Identify active alcohol abuse or dependence.Reduce accessibility and availability of alcohol in the intervened neighborhoods.Reduce environmental neighborhood exposure (public space) to advertising (direct and indirect) that encourages alcohol consumption.

## 2. Materials and Methods

### 2.1. Methods: Participants, Interventions and Outcomes

Study design: This is a quasi-experimental study with a comparison group and pre–post intervention measures (at 24 and 30 months). (Figure 1).

Study setting: Two Barcelona neighborhoods among those who prioritized alcohol-related problems in public audiences will be assigned to the intervention group. As a comparison group, a third neighborhood will be selected to provide similar profiles to the intervention group in terms of socioeconomic characteristics of the neighborhood and heath care data on harmful alcohol consumption (Appendix B, Table A1). 

Eligibility criteria: The study population will be individuals ≥18 years old living in the study neighborhoods. 

Description of the intervention: In January 2019, the first step was to create a community steering group composed of key actors in the study neighborhoods (representatives of local entities, local health services, local police, and technical staff) who co-designed a community intervention based on the evidence [17]. The working group agreed on a name for the group, called AcciOH group, and designed a logo that will appear on all materials designed for the intervention. The following step was to design the activities to be included in the intervention. These actions were defined considering: (a) the available evidence, which points out three effective components in alcohol community interventions: community mobilization, law enforcement, and communication campaigns [17]; (b) a conceptual framework that includes components addressing individual and environmental determinants of harmful alcohol consumption [4]; (c) adaptions to the social determinants and equity axes of the neighborhoods, such as gender and cultural diversity; and (d) being based on the community assets. As this is a multi-component community intervention, the different agents and resources that form part of the local group will lead the activities. The activities will be heterogeneous in terms of the participants targeted, the duration of the activity, the number of participants, the recruitment process, as well as the level of intervention (individual or environmental), the component of the intervention prioritized (community mobilization, reinforcement of regulations, communication campaign), and the behavioral model. Table A2 (Appendix B) presents the details of the potential intervention’s activities based on behavioral models and usual assets in the neighborhoods. No specific activities will be carried out in the comparison group. Population and community groups are expected to continue with their usual activities and programs. 

### 2.2. Methods: Data Collection, Management, and Analysis 

#### 2.2.1. Data Collection

##### Survey

A survey to assess the effects of the multicomponent intervention on individual and personal variables was developed by the research team, based on the Barcelona Health Survey [23], and on available questionnaires for the selected psychosocial variables related to alcohol consumption [24,25,26,27,28]. Baseline data (before implementation) was collected in December 2021 through face-to-face interviews in the street. A sample of 622 subjects (311 in each group) was needed to detect a 10% difference between the IG and the CG (*p* = q = 0,5) main outcomes with an alpha risk of 0.05 and a beta risk of 0.2 in a one-sided contrast, with an estimated loss to follow-up of 10%. Quotas were established for variables of sex (women, men, other), age (20–34 years; 35–54 years, ≥55 years), and nationality (native and immigrant) in each of the neighborhoods (IG and CG) and were applied for the selection of the interviewee (Appendix A for further details. Questionnaire in Spanish).

Trained interviewers will collect the information with a 16-question questionnaire (Appendix A) at two points in time, one prior to the implementation of the intervention (before) and one 24 months after the implementation (after). The interviewers will use tablets with the Gandia Integra Mobinet application downloaded on them. Through this application and by means of a username and password, they will carry out the interviews with the selected interviewees. Those who agree to answer the questionnaire were asked to be contacted again 24 months later to answer the same questionnaire. A shopping voucher worth EUR 100 will be raffled among those who agree to be recontacted. Strategies to ensure the quality and consistency of data included close supervision of interviewers in the field, review of all questionnaires, and re-interviewing of 20% of participants, selected randomly. Inconsistencies during data entry were controlled.

##### Variables

Sociodemographic variables (sex, age, nationality, current employment situation, and level of completed education) were collected following Barcelona Health Survey definitions [23]. The survey outcomes are: (1) risk perception about alcohol consumption; (2) knowledge of the limits of harmful consumption [24,25]; (3) attitudes related to alcohol consumption and self-efficacy to resist peer pressure [26]; and (4) drinking patterns [27] and alcohol use disorders test [28]. 

#### 2.2.2. Data Management

Strategies to ensure the quality and consistency of data included the close supervision of interviewers in the field and double data entry. 

#### 2.2.3. Data Analysis 

A descriptive analysis will be performed to assess changes in the outcomes before and after the intervention. Appropriate parametric and non-parametric statistical tests will be conducted. To compare measures after the intervention and follow-up assessments, mixed models will be applied, and the analysis of covariance (ANOVA) will be applied to control the effects of covariates. Differences between before and after the intervention will be compared with the McNemar test, paired *t*-test, or sign test, according to the type of variable and distribution. A two-sample McNemar test will first be used to analyze differences before and after the intervention for each main outcome. Further analyses will evaluate whether the effects differ by socioeconomic characteristics or other outcomes. Poisson regression models with robust variance [29] will be used for all outcome variables, adjusted to socio-economic characteristics if needed to compare the two groups before and after the intervention, using a difference-in-differences approach.

##### Non-Participant Observations 

Non-participant observations will be performed through the OHCITIES tool [30], which relies on systematic observations for assessing environmental variables related to alcohol availability and accessibility to characterize the urban environment. These observations will be performed at two points in time: one prior to the intervention’s activities implementation (before) and 30 months after the implementation (after). 

The OHCITES instrument includes three main domains [30]: (1) Availability and accessibility of alcohol products in the study neighborhoods: the presence and type of locations where people purchase (off-premises) or consume (on-premises) alcohol. (2) Promotion and advertising elements in the environment: any type of alcoholic advertising or sponsorship elements associated with on-premises and off-premises that can be observed from the street, and any type of promotion in public spaces (including streets, public squares, playgrounds, parks, and other green spaces) beyond the outlet. (3) Signs of alcohol consumption: includes elements wasted in public spaces that allow traces of alcohol consumption to be identified (empty cans, bottles, bottle tops, glasses, etc.). Two observers carried out observations in the public space in standardized census sections: 12 sections in the intervened neighborhoods (IG) and 10 in the comparison group (CG), on 8 days (two weekdays and two weekends) and in three time slots: weekdays (morning-afternoon: 0800–1500 h, and evening: 1830–2100 h) and weekends (1100–1700, 0800–1430, and 1600–2100).

##### Variables 

The outcome variables are: (1) number of on-site establishments for alcohol consumption (bars, restaurants, cafés); (2) number of off-site establishments for alcoholic beverages (supermarkets, grocery shops, etc.); (3) number of establishments selling general products (including alcohol); (4) number of advertisements related to alcoholic beverages/alcohol consumption: advertising or sponsorship announcements in public spaces; (5) number of alcoholic beverages street vendors; and (6) number of bottles, cans, glasses, or other alcohol-related waste in public spaces, beyond on-premises presence, and the presence of people drinking alcohol in public spaces. 

Explanatory variables: neighborhood of residence and days and time slot for data collection.

##### Data Management

Strategies to ensure the quality and consistency of data included the close supervision of interviewers in the field and double data entry. 

##### Data Analysis 

A descriptive analysis will be described summing up all observed elements for the three main domains. Difference-in-differences analysis will be conducted between before and after the intervention, and between the intervention (IC) and comparison group (CG).

## 3. Ethical Considerations

Research ethics approval: The research team is committed to performing this study in accordance with the Good Clinical Practice Guidelines of the Declaration of Helsinki, with its latest updates, including the Oviedo Convention, and the Good Clinical Practice (GCP) guidelines of the International Conference on Harmonization (ICH), and applicable legislation. This protocol has been approved by the Parc de Salut Mar Clinical Research Ethics Committee Board (nº 2022/10429/I).

Consent or assent: Every action undertaken as part of this intervention will respect the necessary ethical standards. Written informed consent will be required and anonymity will be guaranteed.

Confidentiality: In most cases, no individual data will be collected except some general information regarding sociodemographic data needed to analyze the information.

Access to data: When collecting secondary data, databases will be handled in accordance with current data protection laws (Organic Law 3/2018 of 5 December on the Protection of Personal Data and guarantee of digital rights and the GDPR: REGULATION (EU) 2016/679 OF THE EUROPEAN PARLIAMENT AND OF THE COUNCIL of 27 April 2016 on the protection of individuals with regard to the processing of personal data and on the free movement of such data). 

Dissemination policy: Once the evaluation of the intervention is completed and depending on the results found, the comparison group (CG) will be offered the possibility to implement the actions that have proven to be effective. 

## 4. Discussion

### 4.1. The Expected Results of the Study

We can foresee the challenges that we will face in evaluating a multi-component community intervention, as it has been described previously [17]. The available evidence on effective community interventions has methodological weaknesses, mainly in its design, and offers few details of the implementation process of the interventions, which makes it more difficult to determine whether the lack of success is due to the intervention components or to implementation failures.

Knowledge does not usually lead directly to behavior change, but it is a pre-requisite for most of the other determinants, such us risk perception, self-monitoring, behavioral beliefs, perceived norms, and skills [20]. In Spain, in 2020, 50.2% of the population knew the limits for risky alcohol consumption [31]. In our results, we expect individual variables to show significant changes between IG and CG. In the IG, we expect to detect an increase in the knowledge of drinking limits, a change in attitude towards alcohol consumption, and an increased risk perception regarding harmful drinking of more than 10% between both groups after 24–30 months of intervention [32]. However, we know that changes in attitudes and behaviors are slow, and changes in these variables may not be captured in the measurement time available for the study. Regarding the outcomes of environmental variables, we expect to detect in the IG a 10% reduction in alcohol accessibility and availability between IG and CG, 24–30 months after the intervention. 

### 4.2. Strengths and Limitations 

This intervention is unique in our context, which is why the results of the evaluation of its implementation could be useful to assess the effectiveness of community interventions to prevent and reduce harmful alcohol consumption.

The intervention components and its actions are consistent with the SAFER strategy to reduce harmful alcohol consumption [33]. The intervention design is based on the development of a conceptual framework and also on a literature review that has identified the main components that proved to be effective in interventions aimed at reducing harmful alcohol consumption [17]. Another strength is the evaluative design with a comparison group, which will allow us to control potential external factors that could jeopardize the internal validity of the results, as well as the use of observations and quantitative methods.

A potential limitation of the study is the difficulty in capturing individual and environmental changes proposed by the intervention. These are profound changes, which require a long period of time for their incorporation and acceptance. Due to the length of the intervention follow-up (24–30 months), we will not be able to assess its long-term effects. Even though the comparison group strengthens the validity, a not-perfect similarity between the groups may introduce a bias. This will be taken into account in the analysis and discussion.

### 4.3. Dissemination of Results and Future Research

The results of this study will be especially useful to inform community action planning and for the development of practical knowledge on community interventions to reduce problematic alcohol consumption.

Depending on the results of its evaluation, the design of this intervention could be replicated in other similar contexts.

## 5. Conclusions

This study protocol to assess the effects of a multicomponent community health intervention on harmful alcohol consumption at the environmental and population level will provide evidence on effective community health interventions and enable informed decisions on resource allocation and improved public accountability. We will face major challenges in evaluating the individual and environmental effects, but its results on harmful consumption are promising. This protocol could also be used as an implementation guide for studies aimed at reducing harmful consumption in cities with similar characteristics.

## Figures and Tables

**Figure 1 ijerph-19-08746-f001:**
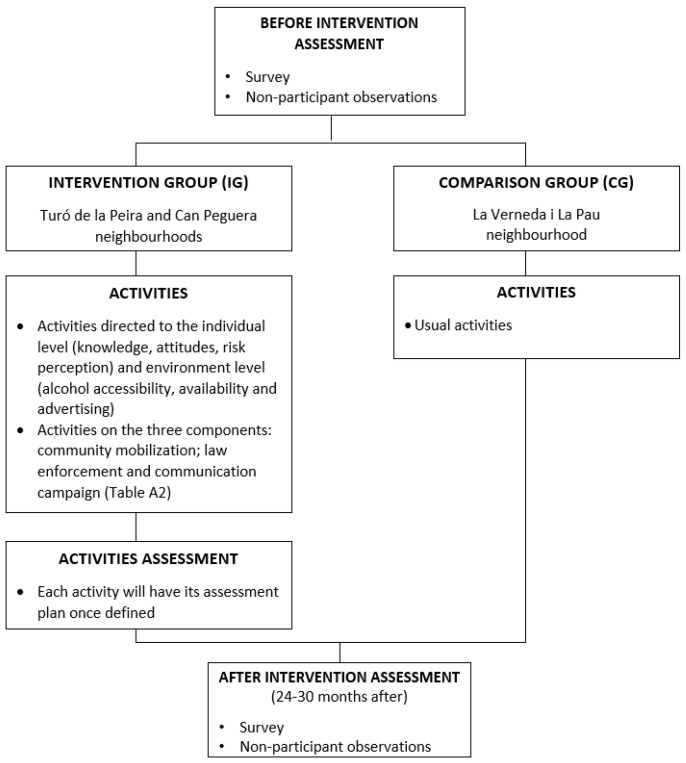
Flow chart of the study design.

## Data Availability

Access to data: When collecting secondary data, databases will be handled in accordance with current data protection laws (Organic Law 3/2018 of 5 December on the Protection of Personal Data and guarantee of digital rights and the GDPR: REGULATION (EU) 2016/679 OF THE EUROPEAN PARLIAMENT AND OF THE COUNCIL of 27 April 2016 on the protection of individuals with regard to the processing of personal data and on the free movement of such data).

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
