# Peer review of "Evaluation of a Community Intervention to Reduce Harmful Alcohol Consumption among Adult Population: A Study Protocol"

_ijerph, 2022, doi:10.3390/ijerph19148746_

Round 1
Reviewer 1 Report
The paper presents a study protocol of a forthcoming evaluation of a community intervention. Even the intervention has not been started yet. When reading the call for the special issue my understanding was that it was not about plans but about concrete results of implementations, effectiveness, or interventions. I doubt if it is worth publishing a plan without knowing anything about its realization.
Apart from the above it is a well-thought-out-study protocol. The aim of the intervention program is to change alcohol related attitudes and knowledge on individual level and reduce alcohol availability on community level. Both the intervention program and its evaluation are well elaborated, relevant, and feasible.
In the discussion chapters authors mention the methodological weaknesses of other community interventions. It would have been useful to learn something more about the lack of success of these programmes in the introduction chapter. It is not clear why they defined the criteria of successfulness in 10% changes in knowledge and attitudes. What this 10% criterion is based on. When evaluating the implementation how they can distinguish real changes in attitudes from the knowledge of the “right” answers (during the program people can learn what is the correct answer, but it doesn’t necessarily mean that their opinion has really changed).
In the introduction they also raise the problem of inequality. It is not clear whether they are planning different interventions for different social groups or how they are going to handle this issue.
It would be also good to discuss, how much the intervention program and the whole study, might be generalized for other drinking cultures?
To sum up, an interesting, well-founded, and well elaborated research protocol, but I’m not convinced that it is worth publishing in this stage.
Author Response
Dear reviewer,
We would like to thank all the reviewers and the editor for their suggestions and contributions. They have certainly helped to improve this article. Below you will find a point-by-point response to each of the comments, pointing out the pages and lines of the manuscript where the change has been made. At the same time, changes have been marked in blue in the new version of the manuscript.
The paper presents a study protocol of a forthcoming evaluation of a community intervention. Even the intervention has not been started yet. When reading the call for the special issue my understanding was that it was not about plans but about concrete results of implementations, effectiveness, or interventions. I doubt if it is worth publishing a plan without knowing anything about its realization.
While it is true that we do not yet have complete concrete data on the implementation or full description of all the actions that make up the intervention, some actions related to the design of the intervention as well as the baseline measures for the evaluation have already been carried out, and the activities are progressively being implemented following the community schedule.
On pages 3-4, lines 130-140, where the intervention is described, we have made changes to the paragraph to make it more understandable. In 2019, the steering group that will work on the intervention was established. Based on the evidence review we have previously done, the steering group defined the objectives of the intervention and some of its activities (table 2). After the pause during the COVID-19 pandemic, in October 2021, the working group agreed on a name for the group, called the AcciOH group, and designed a logo that will appear on all materials designed for the intervention. Recently, in June 2022, we were able to implement a number of activities planned for the community mobilization component, related to the major neighborhood celebrations and the summer annual festival, such as training staff in responsible alcohol dispensing, diversifying alternatives to alcoholic drinks and offering free water during the festivities (described in Table 2). Evaluation has not yet been carried out and is ongoing.
Regarding the evaluation of the intervention, we have been able to make progress on the baseline measures. On page 5, lines 159-169, we have introduced changes and added information that facilitates the understanding of the process. During this time baseline data has been collected (before) for the evaluation. For the questionnaire, which collects data to answer objectives 1-3, data were collected in December 2021. For the non-participant observations (page 6, lines 207-213, which collect data to respond to objectives 4 and 5, baseline data were collected between January and October 2021. We have not yet performed the analysis of these data. At the request of reviewer 2, we attach the supplementary material.
We have also made some changes of the abstract to make it consistent with the rest of the manuscript (see page 1, lines 24 and 27)
Apart from the above it is a well-thought-out-study protocol. The aim of the intervention program is to change alcohol related attitudes and knowledge on individual level and reduce alcohol availability on community level. Both the intervention program and its evaluation are well elaborated, relevant, and feasible.
In the discussion chapters authors mention the methodological weaknesses of other community interventions. It would have been useful to learn something more about the lack of success of these programmes in the introduction chapter.
On page 2, lines 61-68, in the introduction section, we have introduced a detailed description of the methodological weaknesses detected on the literature review we had previously done.
It is not clear why they defined the criteria of successfulness in 10% changes in knowledge and attitudes. What this 10% criterion is based on. When evaluating the implementation how they can distinguish real changes in attitudes from the knowledge of the “right” answers (during the program people can learn what is the correct answer, but it doesn’t necessarily mean that their opinion has really changed).
On page 7, lines 276-288, we have introduced some changes to make clear this point.
We agree with the reviewer that knowledge does not usually lead directly to behaviour change, but it is a prerequisite for most of the other determinants, such as risk perception, self-control, behavioural beliefs, perceived norms and skills. (Bartholomew Eldredge LK, Markham CM, Ruiter RAC, Fernandez ME, Kok G, Parcel G. Planning Health Promotion Programs: An Intervention Mapping Approach. Jossey-Bass Public Health; 2016.).
We have added that, in Spain. Ia 50.2% of the population knows the limits for risky alcohol consumption (Observatorio Español de las Drogas y las Adicciones. Informe 2021. Alcohol, tabaco y drogas ilegales en España. Madrid: Ministerio de Sanidad. Delegación del Gobierno para el Plan Nacional sobre Drogas; 2021. 243 p)
We have also added a quote from a published study showing a 10% increase in some knowledge and attitudes after a communication campaign. Dixon HG, Pratt IS, Scully ML, et al. Using a mass media campaign to raise women’s awareness of the link between alcohol and cancer: cross-sectional pre-intervention and post-intervention evaluation surveys. doi:10.1136/bmjopen-2014-006511.
In the introduction they also raise the problem of inequality. It is not clear whether they are planning different interventions for different social groups or how they are going to handle this issue.
As mentioned in the introduction, on page 2, lines 71-73, the Barcelona Health in the Neighborhoods (BSaB) programme is a strategy to reduce health inequities. The three neighborhoods where the intervention takes place are part of the 23 deprived neighborhoods prioritized by this programme.
Specifically in the activities planned so far, the aim is to include the population of the neighborhood, being aware of aspects related to gender, origin and sex. Whenever possible, gender equity will be incorporated in all actions and the actions will seek to be attractive to each population group in the neighborhood, to incorporate the greatest heterogeneity of the population to participation (by age, origin, etc.) This will be done considering the places/locations where the activities will be carried out, the type of activity offered, the time for the activities, etc. In addition, most of the proposed activities will be free of charge for those who participate.
For the individual evaluation, the questionnaire selects participants based on quotas (according to age, gender and origin), to obtain a sample in accordance with the characteristics of the population of the intervention neighborhoods.
It would be also good to discuss, how much the intervention program and the whole study, might be generalized for other drinking cultures?
Based on the evidence we have been able to review, the three components (community mobilization, communication and law enforcement) that guide the actions that will comprise this intervention may be generalizable to high-income country contexts, where they have proven to be effective.
However, we know that contexts are determinants of drinking culture. As mentioned in the introduction, the available evidence points to the relevance of settings for harmful drinking. The normalization of abusive or socially accepted drinking increases the risk of harmful drinking. Generalizations to other cultures should therefore be made with caution. We cannot guarantee whether outside the context where the intervention takes place, it might have similar effects.
To sum up, an interesting, well-founded, and well elaborated research protocol, but I’m not convinced that it is worth publishing in this stage.
Thank you very much. We have tried to make it clear that the intervention has been designed and is currently being carried out.

Reviewer 2 Report
A community-based approach to alcohol use disorders is a very important way to educate citizens and treat patients. The present study, which seeks to address this issue, is a socially significant and attention-grabbing study.
1. As below, the authors seemingly proposed questionnaire and quota criteria as Supplementary material, which are not shared with me. The author should submit them as well.
"In the case of face-to-face interviews, respondents must reside in the study neighborhoods and will follow the same quota criteria. (Supplementary material for further details. Questionnaire in Spanish)"
"Trained interviewers will collect the information with a 16-question questionnaire (Supplementary material)"
2. The authors identified three common components among the effective interventions in the community setting community mobilization, law enforcement, and media campaigns. The interventions proposed in the manuscript are seemingly based on them, but they don't exactly fix the interventions. So then, they must detect and describe what kind of interventions are actually done in each city.
And also they should declare how they detect the activities like whether a new law is enforced during the research period, how TV or radio broadcasts alcohol events or alcohol products, etc.
Additionally, they won't do any interventions in the control city, but they can't control everything in the city because they are not a decision-maker. Please declare how they confirm that the other group does not have any interventions like theirs.
3. The authors should declare how those cities (Turo de la Peira, Can Peguera, and La Verneda i La Pau) were chosen.
4. According to Table A2, the degree of drug and alcohol harm in the comparison city is less than in intervention cities (index of drug problems, mortality due to adverse reaction to drug use). The authors might want to consider choosing another city like the intervention cities. If they can't change the control city, please declare how they address this issue.
5. Please reasonably declare why they have 2 cities as the intervention cities, not a single city. It is not that they will lack participants in a single city because they need just several hundred people, it means they can have enough participants if they do in one city.
6. As the authors showed, the activities done in each city will be heterogeneous. How do they justify combining the data from both of them? I need you to describe you will at least show data in every outcome in each city, not only combining data.
Author Response
Dear reviewer,
We would like to thank all the reviewers and the editor for their suggestions and contributions. They have certainly helped to improve this article. Below you will find a point-by-point response to each of the comments, pointing out the pages and lines of the manuscript where the change has been made. At the same time, changes have been marked in blue in the new version of the manuscript.
A community-based approach to alcohol use disorders is a very important way to educate citizens and treat patients. The present study, which seeks to address this issue, is a socially significant and attention-grabbing study.
- As below, the authors seemingly proposed questionnaire and quota criteria as Supplementary material, which are not shared with me. The author should submit them as well.
"In the case of face-to-face interviews, respondents must reside in the study neighbourhoods and the same quota criteria will be followed. (Supplementary material for further details. Questionnaire in Spanish)"
"Trained interviewers will collect the information with a 16-question questionnaire (Supplementary material)"
The supplementary material is attached so that it can be assessed by the reviewers.
- The authors identified three common components among the effective interventions in the community setting community mobilization, law enforcement, and media campaigns. The interventions proposed in the manuscript are seemingly based on them, but they don't exactly fix the interventions. So then, they must detect and describe what kind of interventions are actually done in each city.
We would like to clarify, in case it is not clear from the text, that the intervention takes place in 3 neighborhoods of Barcelona, not cities. The neighborhoods Turó de la Peira and Can Peguera are two adjacent neighborhoods that share public resources and technical staff who are part of the steering group. These differences are detailed above. Table A2 details the intervention activities differentiated by each of the three components.
Following the recommendation of reviewer 1, on pages 3 and 4, we have made some changes to the description of the intervention to make it more understandable
And also they should declare how they detect the activities like whether a new law is enforced during the research period, how TV or radio broadcasts alcohol events or alcohol products, etc.
The research team will monitor whether there are any specific campaigns or changes in neighborhood or city regulations that may affect the comparability of the intervention and comparison territories. If changes occur during the intervention period, they will be recorded and described in the subsequent discussion and reflections. However, these territories have been chosen because they share the same characteristics and no specific intervention is planned in any of the three neighborhoods. In addition, the evaluative design with a comparison group will allow assessing the effects of the strategies and policies that can be implemented.
On the other hand, the Barcelona Public Health Agency and its Drug Dependency Prevention and Care Service are among the key agents for advising, recommending or participating in decision-making that may introduce changes in the regulations and standards that affect the city.
Additionally, they won't do any interventions in the control city, but they can't control everything in the city because they are not a decision-maker. Please declare how they confirm that the other group does not have any interventions like theirs.
Indeed, this is beyond our capacity to control. However, our Community Health Service has a team of professionals working in the control group's neighborhood, so if any changes are made or new alcohol interventions are introduced, we would be aware of them and be able to record them for further evaluation.
As we referred on the previous comment, the evaluative design with a comparison group will allow assessing the effects of strategies and policies that can be implemented. We believe the evaluative design of this intervention is another strength of the study, so we have added it in page 8, lines 296-299.
- The authors should declare how those cities (Turo de la Peira, Can Peguera, and La Verneda i La Pau) were chosen.
The criteria used to select the study neighborhoods, health and socio-economic data are described in page 3, lines 123-127. This information is expanded in Table A1, which presents the data for each indicator for each of the three study neighborhoods.
- According to Table A2, the degree of drug and alcohol harm in the comparison city is less than in intervention cities (index of drug problems, mortality due to adverse reaction to drug use). The authors might want to consider choosing another city like the intervention cities. If they can't change the control city, please declare how they address this issue.
For the selection of these neighborhoods, we reviewed the 23 neighborhoods prioritized by the Barcelona Salut als Barris (BSaB) programme of the Barcelona Public Health Agency (Daban et al. Barcelona Salut als Barris: Twelve years’ experience of tackling social health inequalities through community-based interventions. Gac Sanit 2020. https://doi.org/10.1016/j.gaceta.2020.02.007) and within the differences that they may present, La Verneda i la Pau was the most similar neighborhood. Another criterion that was taken into account, although it does not appear in table A1, but is described in page 3, lines 123-124, is that in the health diagnoses carried out in the intervened neighborhoods, problematic alcohol consumption was identified as one of the main health problems in the neighborhoods selected for this intervention.
With regard to the indicators used (Table A1), the biggest differences appear between Can Peguera and the rest of the neighborhoods. This is because Can Peguera, as it shows the socio-demographic data, is a small neighborhood with a small population and this distorts the numbers. The indicator that shows the greatest differences between the Turó de la Peira and La Verneda neighborhoods is mortality due to adverse reactions to opiate use. The rest of the indicators, as well as the socio-demographic data, are quite similar.
From a methodological point of view, the effect will be measured from the differences between before and after among the comparison and intervention groups.
- Please reasonably declare why they have 2 cities as the intervention cities, not a single city. It is not that they will lack participants in a single city because they need just several hundred people, it means they can have enough participants if they do in one city.
Indeed, the criterion for including two intervention neighborhoods and one control neighborhood is not due to the number of participants. Turó de la Peira and Can Peguera are adjoining neighborhoods, separated by a street. The administrative division does not reflect the reality of the neighborhoods, which share political leadership, resources and public facilities, as well as the professionals working in the area. At the time we were selecting neighborhoods for the intervention, it was decided to incorporate Can Peguera, which had also prioritized problematic alcohol consumption in the health diagnosis.
- As the authors showed, the activities done in each city will be heterogeneous. How do they justify combining the data from both of them? I need you to describe you will at least show data in every outcome in each city, not only combining data.
We are not sure if we understand this comment. We hope that the doubt has been clarified by the above answers.
The inhabitants of the two neighbourhoods, which are adjacent, will be able to participate in the various activities that will take place in the intervention group. In the comparison group, no similar activities will take place.
In the data analysis, we have improved the explanation of the description. The data will be analysed to estimate treatment effects by comparing the change (difference) in the differences in observed outcomes between the treatment and control groups, over the pre- and post-treatment periods, using a difference-in-difference approach.
On page 5, line 200-201, we have added a sentence to make it clearer.
Thank you very much. We have tried to incorporate all your valuable comments.

Round 2
Reviewer 1 Report
The authors went through all the comments and questions. They have clarified the criticized parts of the paper and answered the questions correctly. The article has become more understandable now.
Author Response
Dear reviewer,
Once again, we would like to thank you and the editor for your suggestions and contributions to the manuscript. They have certainly helped to improve our work.
Best regards,
Victoria